# Association of microtubule destabilization with platelet yields in terminally differentiating hiPSC-derived megakaryocyte lines

Emiri Nakamura[1], Yasuo Harada[1¤a], Trevor Bingham[2], Christian Skorik[2¤b], Anjali Jha[2¤c], John Atwater[2¤d], Natsumi Higashi[1], Kosuke Fujio[1¤a], Mariko Ishiguro[1], Haruki Okamoto[1], Leonard I. Zon[2], George Q. Daley[2], Andrew L. Frelinger[3], Koji Eto[1], Thorsten M. Schlaeger[2]*

1 Center for iPS Cell Research and Application, Kyoto University, Kyoto, Japan, 2 Boston Children's Hospital Stem Cell Program and Harvard Medical School, Boston, Massachusetts, United States of America, 3 Center for Platelet Research Studies at Boston Children's Hospital, Boston, Massachusetts, United States of America

¤a Current Address: Otsuka Pharmaceutical Co., Ltd., Osaka, Japan
¤b Current Address: Stem Cell Technologies, Cambridge, Massachusetts, United States of America
¤c Current Address: Department on Epidemiology, Dartmouth Geisel School of Medicine, Hanover, New Hampshire, United States of America
¤d Current Address: Vilcek Institute of Graduate Biomedical Sciences, Grossman School of Medicine, New York University, New York, New York, United States of America
* thorsten.schlaeger@childrens.harvard.edu

## Abstract

Millions of platelet units are needed each year to manage thrombocytopenia and other conditions linked to excessive bleeding. These life-saving treatments still depend entirely on donated platelets, despite the numerous shortcomings associated with them, such as limited shelf life, supply shortages, unpredictable functionality, potential for infection, as well as immune-incompatibility issues. These challenges could be overcome with universal donor platelets generated from human induced pluripotent stem cell (hiPSC)-derived megakaryocytes (MKs). We recently developed expandable hiPSC-derived megakaryocytic cell lines (imMKCLs) as a potentially unlimited source for platelet production. imMKCL-derived platelets are functional and have already been tested in patients. In this study, we demonstrate through single-cell time-course imaging that imMKCL maturation is heterogeneous and asynchronous, with only a few imMKCLs generating platelets at any given time under static culture conditions. Using a chemical screen, we identify microtubule (MT) destabilizing agents, including vincristine (VCR), as promising hits, with a larger proportion of VCR-exposed imMKCLs developing proplatelet extensions and more platelets being produced per imMKCL. VCR use reduces the MT content of imMKCLs and results in the production of platelets with a diminished peripheral MT ring structure. Nevertheless, these platelets are functional, as evidenced by their normal response to agonists, their ability to attach to and spread on fibrinogen-coated surfaces, and their capacity to restore hemostasis *in vivo*. Interestingly, we also observed a

**Data availability statement:** All relevant data are within the paper and its Supporting Information files.

**Funding:** "This work was made possible by the generous support of the Boston Children's Hospital Stem Cell Program, the NIDDK (U54DK110805), Megakaryon Inc., and the Massachusetts Life Science Center (Research Infrastructure Program, 2013 project 'Children's Center for Cell Therapy' and 2024 project 'Stem Cell Imaging and Therapeutics Center'). This work was supported in part by the Core research center for next-generation medicine utilizing Cell and gene therapy grant (23bm1323001h0001 to K.E.) from AMED, the CiRA Foundation Fund (to K.E.), a grant-in-aid for scientific research (KIBAN S, 21H05047, to K.E) from the Japan Society for the Promotion of Science (JSPS), and collaborating project funding from Megakaryon Co. Ltd. (to K.E., A. F., and T. S.). The funders had no role in study design, data collection and analysis, decision to publish, or preparation of the manuscript."

**Competing interests:** "K.E. was a founder of Megakaryon and has not stock currently and received research funding from Megakaryon, Otsuka Pharmaceutical, and Kyoto Manufacturing Co. T.S. received research funding from Megakaryon Inc. Y.H. and K.F. are currently employed by Otsuka Pharmaceutical CO., Ltd., Osaka, Japan. This does not alter our adherence to PLOS ONE policies on sharing data and materials. The remaining authors declare no competing financial interests."

negative correlation between the MT content of imMKCLs and platelet yields when we compared imMKCLs differentiated under static conditions (MT$^{high}$, low yield) to our turbulence-optimized VerMES™ bioreactor (MT$^{low}$, high yield). Taken together, our findings highlight the importance of MT dynamics in megakaryocyte biology, provide a possible explanation for the still poorly understood link between vinca alkaloid *in vivo* use and thrombocytosis, and bring us closer to realizing the clinical potential of affordable, off-the-shelf hiPSC-derived platelets.

## Introduction

Bone marrow MKs produce anucleate platelets that play critical roles in hemostasis, blood coagulation, vessel integrity, and immunity [1,2]. Over 2 million units of platelets are transfused annually in the U.S. alone to manage thrombocytopenia [3], but donor platelets come with major limitations, including a requirement for room temperature storage that limits shelf-life to 5 days and exacerbates the risk of platelet shortages and transfusion-transmitted sepsis [4]. In addition, patients may become refractory to transfusion of allogeneic platelet-rich plasma, making its use ineffective or even detrimental [5,6].

An attractive alternative would be the large-scale manufacture of platelets of consistent quality from universally histo-compatible human induced pluripotent stem cells (hiPSCs), a renewable cell source that could potentially alleviate or even eliminate the continuous need for human platelet donors [7–11]. Still, mass production of platelets from hiPSCs faces its own challenges. During terminal differentiation, MKs become polyploid, increase their size, produce pro-platelet like extensions, and ultimately release (pro)-platelets. This process is often very inefficient *in vitro*, particularly when compared to their *in vivo* counterparts [7,12–15]: *in vitro*, most MKs differentiate asynchronously, fail to become highly polyploid, and often only release a few functional pro-platelets [8,10,16–18]. Furthermore, generating large quantities of platelet-producing MKs by starting each production run from hiPSCs is a slow and complex multi-step process that requires many different medias and culture platforms, posing significant challenges for large-scale production and clinical translation.

The need to produce new MKs from hiPSCs during each platelet production run can be avoided by conditionally immortalizing hiPSC-derived megakaryocytic cells using doxycycline (DOX)-dependent expression of c-MYC, BMI1, and BCL-XL. The resulting stable cell lines, called imMKCLs, can be expanded exponentially as not yet fully mature (diploid, CD42b-low) megakaryocytic cells in simple, low-maintenance suspension culture systems [8]. Importantly, expanded imMKCLs retain their ability to undergo terminal differentiation and platelet biogenesis upon DOX removal in a process that closely resembles normal MK biology: during the first phase (days 0–3 of terminal differentiation), imMKCLs stop dividing, grow in size, undergo polyploidization, and increase their membrane content. During the subsequent 24-72h (days 4–6), they produce proplatelet extensions and shed functional (pro-)platelet-like particles [8,19].

Here, we report that this process is heterogeneous and asynchronous, with only a minority of imMKCLs producing pro-platelet-like extensions and platelet-like particles at any given time. Through the first-ever high-content confocal imaging-based chemical genetics screen for proplatelet formation promoting agents we identify microtubule (MT) destabilizing agents, including vinca alkaloids vincristine

(VCR) and vinblastine, as novel and potent inducers of imMKCL terminal differentiation. Interestingly, we observed a similar negative correlation between platelet yields and megakaryocytic MT content when we compared imMKCLs differentiated in static culture (low platelet productivity) to those made using our turbulence-based VerMES™ bioreactor [19] (high platelet yield).

## Materials and methods

### Source of imMKCLs

imMKCL CL17 (subclone of CL-7) was established following the method as stated our previous publication [8,19] for establishment of CL-7.

### Culture of imMKCLs

For exponential cell expansion, imMKCL clones SeV2-7 (CL-7) or SeV2-17 (CL-17) were cultured in 'DOX-ON' MK growth medium containing doxycycline (DOX) to maintain expression of the conditionally immortalizing transgenes, c-MYC, BMI1, and BCL-XL. imMKCLs were cultured in a humidified incubator at 37°C and 5% $CO_2$ and passaged to maintain a density of $1 \times 10^5$ to $1 \times 10^6$ cells/mL in this medium [19]. The six-day maturation process (D0-D6) was initiated by seeding imMKCLs at $1 \times 10^5$ imMKCLs/mL in 'DOX-OFF' terminal differentiation/maturation lacking DOX and KP457 [19]. VCR was added to the DOX-OFF medium either on D0 or on D3 of the terminal maturation phase. Differentiation cultures were performed in static (non-shaking) 10 cm tissue culture dishes (Thermo, #150466), 125–500 mL Corning Erlenmeyer cell culture flasks (#431143, #431144 and #431145; Sigma-Aldrich) in orbital shaking conditions (100 rpm), or in the VerMES™ bioreactor.

### Reagents

The drug libraries used for high throughput screen are listed in S1 Table. All the candidate compounds/drugs, antibodies and physiological agonists that were used for downstream characterization and validation experiments are listed in S2 Table.

### Chemical genetics high throughput screen (HTS)

Dispenses and pipetting steps were carried out using Biotek Multiflo FX (1µl increment peristaltic cassette) and Apricot PP384-M (15µl tips) devices. Chemical libraries (S1 Table) were reformatted (240 compounds/plate in wells C03-N22, 2µl/well) and stored as single-use aliquots at −20°C under dry nitrogen in sealed 384w plates (AB-1056, Thermo). Each compound was tested at 4 different dilutions (1:500–1:62500). D0 imMKCLs were stained with PKH26 (5µl/1M cells), Hoechst-33342 (25 µg/ml) and Sytox Green (50nM), cultured in differentiation medium (supplemented with AMG9810 to reduce background signal) for three days, resuspended in fresh differentiation medium, and plated into 1536-well plates (Greiner #789866) at 7µl (250 cells)/well. After addition of diluted compounds (2x concentrated, 7µl/well) from our library of bioactive compounds and chemical probes, the assay plates were incubated in a humidified incubator at 37°C and 5% CO2 for 72h with MicroClime lids (Labcyte #LLS0310). On D6 the plates were imaged on a Yokogawa CV7000 confocal imager (10x lens, CH1 = 405nm excitation, 445/45nm acquisition, CH2 = 488nm excitation, 525/50nm acquisition, CH3 = 561nm excitation, 600/37nm acquisition; 12µm Z-stack). Z-stacks were collapsed (sum) followed by image processing and analysis using FIJI (version 1.52p). All processing and analysis scripts are included in the Supplemental Information section, along with spreadsheets with the data for all plotted figures. Processing included background

subtraction, removal of noise and outlier pixels, log-transformation (except for Hoechst), cropping, identification of maxima (to estimate the number of nuclei and particles) and cells (using size and intensity thresholds), removal of cell-free areas, and RGB-pseudo-coloring. Similar results were obtained when platelet-sized particles were identified based on absence of DNA signal and a membrane signal above background that covered 7–200 pixels ($\sim$3–85$\mu m^2$). Per-well-averages of cell areas, perimeters, and other high-content measurements were calculated. The chemical libraries used in this screen are summarized in S1 Table and a compound frequency distribution plot is provided in S1 Fig panel D.

### Single cell time course imaging

imMKCLs were pre-differentiated in bulk to day 3 followed by DiD membrane dye staining and plating at low density into Kuraray/Elplasia 384-well microwell-patterned imaging plates (Corning, 4447) that contain 200x200x100$\mu$m (WxDxH) microwells that keep suspension culture cells contained to allow long-term imaging under static (non-shaking) culture conditions. Time-course imaging was performed on a Yokogawa Cell Voyager (CV7000) imager with environmental control using a 20x lens (4 fields of view per well, each capturing $\sim$12 micro-wells) or a 4x lens (capturing an entire well containing $\sim$625 usable microwells) starting on day 3.5 of terminal differentiation. Microwells containing more than one imMKCL were excluded and no imMKCL cell divisions were detected during the observation period. Z-stacks were collapsed and analyzed in FIJI (version 1.52p). All scripts and data tables are included in the Supporting Information section.

### Secondary image-based validation screen

Chemicals identified from the high-throughput chemical screen were dispensed using a Formulatrix Mantis™ microfluidic liquid handler. Cell and media dispensation steps were carried out using Biotek Multiflo FX (5 µL peristaltic cassette) and Apricot PP-384-M (50 µL tips) devices, respectively. Each compound was tested at 23 different concentrations (5.0E-5 to 2.4E-11 nM). On D0 imMKCLs were stained with PKH26 (5 µL/$10^6$ cells) and Hoechst-33342 (25 µg/mL) and cultured in differentiation medium for 3 days, resuspended in fresh differentiation medium, and plated into 384-well plates (Greiner #781097) at 45 µL (500 cells)/well. After addition of diluted compounds (2x concentrated, 45 µL/well) the assay plates were incubated in a humidified incubator at 37°C and 5% CO2 for 72h. On D6 the plates were imaged on a Yokogawa CV7000 confocal imager (20x lens, CH1 = 405nm excitation, 445/45nm acquisition, CH3 = 561nm excitation, 600/37nm acquisition; 10µm Z-stack). Z-stacks were collapsed (sum) followed by background subtraction, log-transformation, and pseudo-coloring using FIJI (version 1.52). The total number of nucleated cells and the number of nucleated cells bearing extensive proplatelet-like extensions were counted by two individuals who were blinded to the treatment conditions (eight fields of view per condition were counted). All scripts and data tables are included in the Supporting Information section.

### Flow cytometric analyses

Flow cytometric analysis was performed on a Becton Dickinson FACSLyric cytometer. imMKCL-derived platelets (iPSC-PLTs) were stained with the following reagents: anti-hCD41a-APC (#303710), anti-hCD42b-PE (#303906), anti-hCD62P-BV421(#304926) antibodies from Biolegend, FITC-PAC-1 (#340507) antibody, FITC-Annexin V (#556419) from BD. Platelet activation was measured by staining for PAC-1 and CD62P epitopes before and after stimulation with a mixture of 40µM Trap6 and 100µM ADP or with 200nM PMA. Annexin V binding was measured without stimulation or after addition of either 20 µM ionomycin or 25 µM EDTA. Platelet count was measured using Trucount Tubes (#340334, BD Biosciences). The final platelet count per MK was determined by total CD41$^+$/CD42b$^+$ platelets counted on day 6 divided by total imMKCLs seeded on day 0. For ploidy analysis, day 4 imMKCLs were sampled and stained using Leucocount reagent and Trucount tubes (Becton Dickinson, #662415). For MK Cell size analysis, we stained day 4 imMKCLs with anti-hCD41 and anti-hCD42b antibodies in Trucount tubes. We set a acquisition threshold for>5K on the FSC-A axis to eliminate platelet-sized particles and further gated for MK sized cells>50K to isolate MK sized cells. The FCS files of the shown flow cytometry plots are included in the Supporting Information section.

### *In vivo* hemostasis and platelet persistence assay—preparation of platelet concentrate

imMKCLs were cultured in 3 separate 2.4L VerMES™ bioreactors[19]: control, 10nM VCR, or 10µM VCR for 6 days where VCR was added on day 3 of culture. Platelet containing culture suspensions were filtered in a hollow fiber cylinder and concentrated using ACP centrifugation system (HAEMONETICS ACP*215)[19]. Harvested platelets were stored and further concentrated in PLT storage solution with bicarbonated Ringer's solution with 10% ACD-A solution and 2.5% Albumin.

### *In vivo* hemostasis and platelet persistence assay—hemostasis assay

Animals were kept in groups under 12-hour light/12-hour dark cycle with constant access to food and water under pathogen free conditions with ventilating cage covers. Male NSGS-SGM3 mice were bred in-house in the animal facility of the Center for iPS Cell Research and Application, Kyoto University and obtained 3 weeks after birth and used for experiments at 8 weeks. All animal experiments conducted in Kyoto University followed all guidelines approved by the Ethical Committee of Kyoto University. Thrombocytopenia was induced in 8-week-old male NSGS-SGM3 mice by irradiation (2.4Gy) 9 days prior and further by injection of anti-mCD42b antibody (50ng/g) 2 hours prior to conducting the assay. $4 \times 10^8$ (100 µL of $4 \times 10^9$ platelets/mL) or vehicle were injected through the tail vein of mice followed by anesthesia by isoflurane. Hemostasis was then assessed by observing tail bleeding time after tail incision at 2.5 cm from tail end using 23G needle.

### *In vivo* hemostasis and platelet persistence assay—platelet persistence assay

Irradiated mice were administered with $4 \times 10^8$ platelets or vehicle. Peripheral blood samples were collected from the external jugular vein at 30 min, 1 h, 2 h, 4 h, 6 h, and 24 h following injection. The blood samples were stained with anti-hCD41-APC and anti-mCD41-PE-Cy7, and the number of human platelets was measured using FACSLyric, counted as hCD41+/mCD41- population.

### Microtubule staining assay

Microtubules were stained using Spirochrome SPY650-Tubulin (#CY-SC503, Cytoskeleton Inc.) along with Plasmem Bright Green (#P504, Dojindo Laboratories), Spirochrome SPY555-FastAct (#CY-SC202, Cytoskeleton Inc.), Hoechst 33342 (#62249,Thermo), and MitoBright LT Green (#MT10, Dojindo Laboratories). The cultured cell suspension was transferred to an ibiTreat 60µ-dish 35 mm quad (#80416, ibidi) with the staining reagents and incubated in a humidified 37°C, 5% $CO_2$ incubator for 2 hours in the dark. Confocal images were obtained using LSM900 Airyscan confocal microscope with a 63x/1.40 numeric aperture oil immersion objective (Carl Zeiss).

### Platelet spreading assay

ibiTreat 60µ-dish 35 mm quad (#80416, ibidi) were coated with 100 µg/mL human fibrinogen (Sigma Aldrich, #341576) and blocked with 1% BSA. Human PLTs were isolated from whole blood samples using centrifugation and washed with PLT storage solution with bicarbonated Ringer's solution with 10% ACD-A solution and 2.5% Albumin supplemented with $PGE_1$ (PLT buffer). iPSC-PLTs were collected on day 6 and isolated by centrifugation and washed with PLT buffer. Platelets were plated ($2 \times 10^6$ cells per well) on to fibrinogen coated surface with or without 200µM ADP and 40µM Trap-6 in Tyrode's buffer. The cells were then stained with anti-hCD41-APC antibody and SPY555-FastAct and imaged using LSM900 Airyscan confocal microscope with a 63x/1.40 numeric aperture oil immersion objective (Carl Zeiss).

### Statistical analysis

All statistical analysis was conducted using two tailed t-test assuming equal variances (Student's t-test), n = 3 experiments. If t-test was not used, the statistical test is stated in the figure legends.

## Results

### Asynchronous and variable platelet production by imMKCLs under static culture conditions

Down-regulation of the DOX-dependent immortalizing transgenes (c-MYC, BMI1, BCL-XL) triggers irreversible terminal differentiation of imMKCLs. During this process, the cells stop dividing, become larger, increase their membrane content, become polyploid, up-regulate maturation markers, and ultimately produce proplatelet extensions and release proplatelet-like particles [8,19] (Fig 1A). Terminal differentiation completes within six days, which is considerably faster than the time required to produce platelets from hiPSCs using protocols that do not involve conditional immortalization of mega-karyocytic cells [20–23]. Adding small molecules such as the ROCK inhibitor Y39983, the AhR inhibitor GNF351, and the ADAM17 inhibitor KP457 can improve MK maturation as well as the yield and quality of the produced platelets; how-ever, even with these improvements, the average imMKCL ploidy levels and number of platelets produced per imMKCL [8,19,24] remained significantly below those ascribed to MKs *in vivo* [7,15].

We performed single-cell time course confocal imaging to determine if platelet production was uniformly low for all imMKCLs or if individual cells differed in their ability to produce platelet-sized particles under static (non-shaking) culture conditions. The observed mean diameter of individual imMKCLs captured in microwells for time-course imaging was 22.6μm, consistent with the 20–30μm size reported previously for human pluripotent stem cell derived mature MKs and

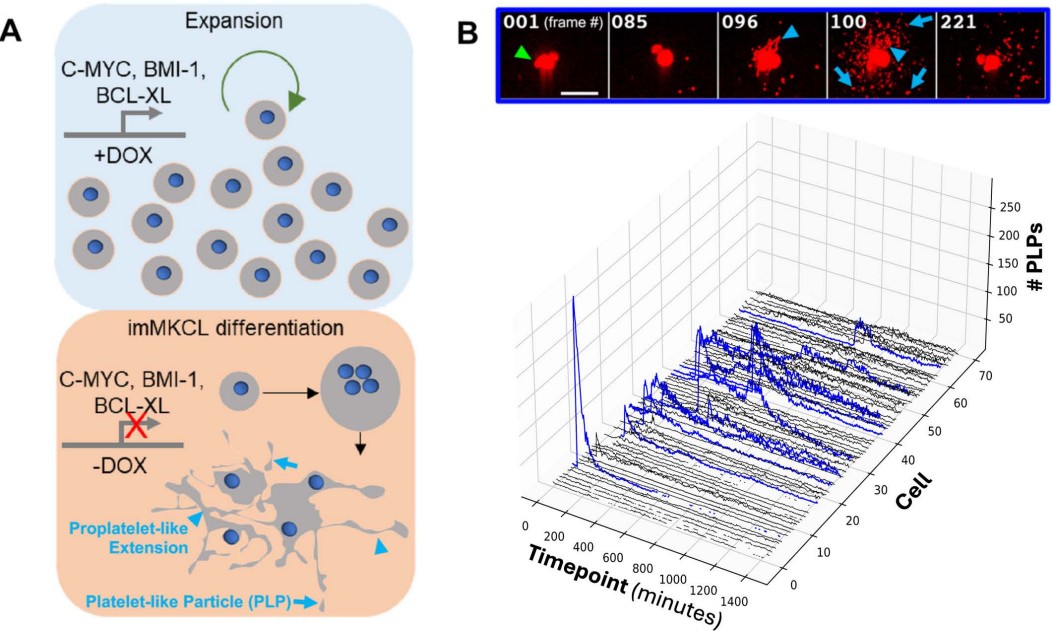

**Fig 1. Single-cell time-lapse imaging reveals heterogeneous and asynchronous platelet biogenesis/release by terminally differentiating imMKCLs under static *in vitro* culture conditions.** (A) (Top Panel) Schematic diagram showing imMKCL expansion, facilitated by DOX-dependent expression of c-MYC, BMI1, and BCL-XL. (Bottom Panel) Schematic diagram showing imMKCL terminal differentiation triggered by DOX removal (down-regulation of exogenous c-MYC/BMI1/BCL-XL) that leads to cell enlargement, polyploidization, formation of proplatelet extension (blue arrowheads), and release of (pro)platelet-like particles (PLPs, blue arrows). (B) Confocal time-lapse imaging of individual imMKCLs undergoing terminal differentiation in static *in vitro* suspension culture conditions. Timepoint 0 corresponds to day 3.5 of differentiation. Individual cells reside in 200x200x100μm microwells patterned at the bottom of the imaging plate. Plotting of the number of proplatelet-like extensions and released platelet-like particles (blue arrowheads and arrows, estimated by counting membrane dye fluorescence intensity maxima near imMKCLs) demonstrates the asynchronous and heterogeneous nature of imMKCLs maturation and (pro)platelet formation/release. Blue lines represent rare cells that produced >50 particles). The post-peak decline in the number of platelet-like participles is due to the slow movement of released particles out of the microwell. Gaps (incomplete lines) represent lack of data for the corresponding timepoint due to autofocus failure. Top: representative individual frames of an imaging dataset from an individual microwell. Scale bar = 50μm.

imMKCLs [8,25]. Like primary human fetal/newborn bone marrow MKs [26], mature imMKCLs are therefore, on average, somewhat smaller than their adult bone marrow counterparts that can reach diameters larger than 40μm [26–28].

These experiments also revealed that most imMKCLs only produced a relatively small number of proplatelet-like particles (PLPs) during the observation period (Fig 1B). Still, a few imMKCLs produced much larger quantities (Fig 1B, S1 Movie): about 1% produced >200 PLPs, 11% produced > 50 PLPs, and over 50% of imMKCLs produced fewer than 10 PLPs. Furthermore, we observed a positive correlation between quantitative markers of imMKCL maturity (cell size and membrane content) and platelet productivity (number of PLPs released), with Pearson correlation coefficients of r = 0.290 (p = 0.004) for cell size and r = 0.296 (p = 0.003) for membrane content (Student t-test), suggesting that more mature imMK-CLs are able to produce and release a larger number of PLPs. Binning the results by size confirmed that large imMKCLs (>26μm; see also S1 Fig panel A) produced more PLPs per cell than small (<15μm) imMKCLs (28.0 vs 12.5 PLPs per cell). We also subcloned the heterogeneous imMKCL line to test if the observed heterogeneity was due to (epi)genetic drift of the culture. However, imMKCLs from a subclone with improved overall platelet production still exhibited a wide range or PLP productivity (S1 Fig panel B), suggesting that cell-external factors such as missing signals or biomechanical forces rather than genetic or epigenetic drift are chiefly responsible for heterogeneities observed under static culture conditions.

## High-content chemical genetics screen for compounds that promote production of PLPs

These observations motivated us to perform an imMKCL based chemical genetics screen to identify platelet biogenesis boosting compounds and pathways. The screen was designed to assess changes in the size (area) and shape (isoperimetric ratio) of compound-treated imMKCLs, parameters expected to increase with maturation and the formation of proplatelet extensions. On day 3 (D3) of differentiation, imMKCLs were labelled with dyes (S1 Fig, panel C, E) to facilitate identification of nuclei, membranes and anucleate platelet-like particles. The stained cells were then cultured for an additional three days in 1536-well plates in the presence of chemical library compounds followed by end-point confocal imaging on D6 (see Fig 2A, B). A total of 15360 compound wells (comprising 3730 unique compounds, each tested at four concentrations; see also S1 Fig, panel D) were screened and the resulting images analyzed to quantify the number, size, and shape of the imMKCLs and PLP (S1 Fig, panels E, F). Overall, imMKCLs differentiated in 1536-well plates exhibited typical characteristics of terminal maturation, including polyploidization, cell growth, and increased membrane content (S1 Fig, panel F), confirming that the assay conditions provided a physiological setting conducive to platelet formation.

Datapoints representing the well averages of cell size and cell shape parameters were gated to identify compounds that induced a cell spreading phenotype (isometric increase in cell size, green gate in Fig 2B) or a phenotype consistent with enhanced formation of proplatelet extensions (increased isoperimetric ratio; red gate). This analysis revealed two small sets of wells that each contained multiple hits (Fig 2B, C). Eleven of the 13 spreading-phenotype hit wells contained PKC agonists, representing a 2,168-fold enrichment of this target class in the green hit gate. Since PKC activation is known to induce MK spreading [29], the strong and specific enrichment of PKC agonists in this gate underscores the physiological behavior of the terminally differentiating imMKCLs under these culture conditions and highlights the specificity and robustness of the assay and analysis. The 51 hit wells in the pro-platelet-like extension phenotype gate included 11 compounds with >1 hit well per compound. Among these, no pathway was represented by more than one compound, except for microtubule destabilizers that, strikingly, were represented by four distinct compounds, corresponding to 13 independent hit wells (see Fig 2C), representing a 280-fold enrichment of this target class.

## Treatment of terminally differentiating imMKCLs with vinca alkaloids promotes formation of pro-platelet-like extensions and platelet production

We prioritized analysis of vinca alkaloids from among the classes of MT destabilizers because they were the only structurally-related hit compounds in this group and because numerous animal and human studies have linked *in vivo* use

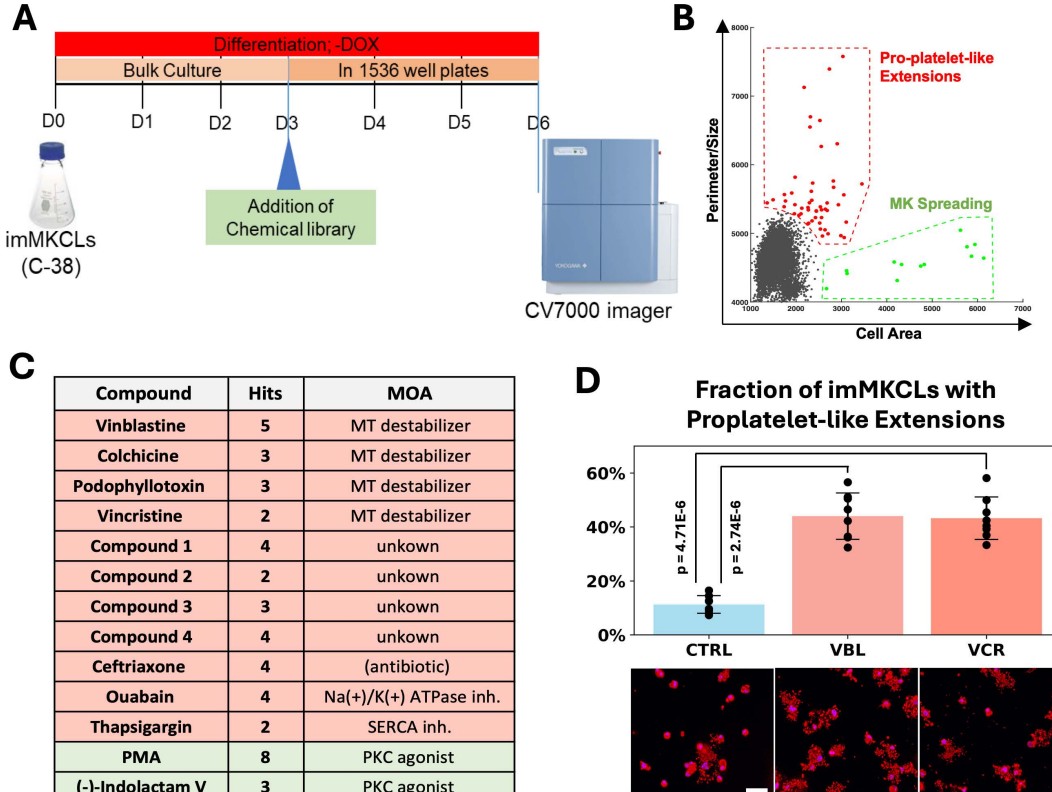

**Fig 2. Identification of proplatelet-extensions inducing compounds through a high-throughput/high-content confocal imaging based chemical genetics screen.** (A) Chemical genetics screening strategy. imMKCLs were expanded, stained (D0) with Sytox Green (a non-permeable dead cell nuclear dye that is used here as a cell mask dye due to its ability to also stain the surface of imMKCLs), PKH26 (a cell-permeable lipid membrane dye that stains outer and internal lipids and membranes) and Hoechst (cell-permeable DNA dye). The cells were differentiated in bulk until D3, then plated at 250 cells/well into 1536w plates wells and mixed with chemical library compounds. Confocal images were acquired for all wells on D6 using a CV7000 high content imager using a 10x lens. (B) High-content image analysis data was generated using image processing and measurement scripts developed and executed with FIJI/ImageJ. For each well, the average isoperimetric ratio (ratio of the average cell perimeter to average radius) of large, nucleated cells (imMKCLs) was plotted against their average size to reveal wells with phenotypic changes expected to occur during enhanced terminal differentiation/platelet biogenesis, such as cell growth/spreading (isometric increase in cell size) or the formation of irregular cell protrusions indicative of the formation of proplatelet extensions (increase in the isoperimetric ratio). Wells exhibiting primarily a cell growth/spreading phenotype (increase cell size with few protruding cell extensions) were gated as 'MK spreading'-phenotype hits (green gate) and wells showing increased cell perimeter to size ratios were gated as 'Pro-platelet-like Extensions'-phenotype hits (red gate). (C) Table with all compounds that were represented more than once in these hit gates. Compounds present N times (1≤N≤5; see also S1 Fig panel D) in the library were screened in 4*N wells (each compound was screened at 4 different concentrations). (D) Secondary hit validation screen. Control (DMSO) or hit compounds vincristine or vinblastine were added on D3 of maturation of imMKCLs stained with the nuclear dye Hoechst (blue) and the membrane dye PKH26 (red) followed by confocal imaging (20x lens). The frequencies of imMKCLs bearing proplatelet-like extensions were counted in a treatment-blinded manner (error bars represent σ, p-values were calculated using Student's t-test).

of vinca alkaloids to thrombocytosis [30–34]. PKC agonists, thapsigargin, and ouabain were from further analysis due to their propensity to induce activation-like phenotypes [35–37] or apoptosis [38]. Likewise, ceftriaxone and compounds 1–4 were deprioritized because their interactions with mammalian targets are unclear.

Consistent with the chemical screening results, imaging-based hit compound validation and titration studies confirmed that vinblastine (VBL) and VCR caused a significant increase in the fraction of proplatelet-bearing imMKCLs (Fig 2D), with vinca alkaloid exposed cells producing larger and more complex proplatelet extensions (S2 Fig panel A) and more PLPs per cell (S2 Fig panel B), at concentrations as low as 6 nM (Z-Score threshold of 6; S2 Fig, panel C). Delaying VCR

addition to day 5 strongly reduced its proplatelet formation boosting effect (S2 Fig panel D). Interestingly, VCR addition on day 3 increased neither the fraction of polyploid cells (S3 Fig panels A,B) nor the size of the imMKCLs (S3 Fig panels C,D), suggesting that it promotes platelet biogenesis without augmenting these conventional markers of MK maturity, much like human fetal MKs that also remain smaller and less polyploid compare to their adult counterpart [26]. Taken together, these results demonstrate the effectiveness of our chemical genetics screening approach in identifying compounds – particularly MT-destabilizing vinca alkaloids – that promote imMKCL platelet biogenesis.

**Turbulence-inducing culture system reduces MT staining in maturing imMKCLs**

To investigate whether MT depletion/reduction also correlates with platelet yields in other imMKCLs terminal differentiation culture systems, we utilized our VerMES™ *in vitro* bioreactor (Fig 3A) that generates optimal turbulent flow conditions shown to boost platelet production compared to conventional culture systems [19]. We analyzed imMKCLs as well as the per-MK platelet yields from three culture systems: (1) static dish culture without any turbulent flow, (2) shaker flask cultures in which some turbulent flow is present, and (3) the turbulent flow optimized VerMES™. As reported previously [19], the yield of CD41+/CD42b+ (pro)platelets (Fig 3B; see also S3 Fig panel E) increased dramatically when turbulent flow conditions are present, particularly with the VerMES™ bioreactor (Fig 3C). Next, we compared the MT content (Fig 3D) of imMKCLs during maturation culture performed using the same three culture systems (dish, flask, and VerMES™). Strikingly, in addition to the expected much-improved per-MK platelet yields (Fig 3C) we found that imMKCLs sampled from shaker flasks or the VerMES™ bioreactor exhibit reduced MT staining in terminally maturing imMKCLs compared to those sampled from static culture conditions, especially after day 3 (Fig 3D; see also S3 Fig panel F). Since exposure to turbulent flow conditions appears to at least partially phenocopy the expected effect of VCR addition, we next tested if VCR use could further boost PLT yields under these culture conditions. Indeed, addition of VCR on day 3 of imMKCL terminal differentiation cultures performed using shaker flasks or the VerMES™ system resulted in statistically significant increases, boosting PLT yields by ~90% (flask, VCR *vs* control) and ~40% (VerMES™, VCR *vs* control) (Fig 3E).

Our data revealed an intriguing inverse correlation between the final platelet yields (day 6) and the MT content of imMKCLs beginning on day 3 of differentiation (Fig 3C, D, E). Notably, the timing of megakaryocytic MT depletion coincides with the point at which VCR is introduced in our chemical genetics screen and subsequent hit validation experiments (Fig 2B,D, and S2 Fig panels A, B, C). In contrast, when VCR was added on day 5 (instead of day 3) to imMKCLs differentiating in dishes, it markedly reduced the frequency of imMKCLs forming proplatelet-extensions on day 6 (S2 Fig panel D). Likewise, adding VCR earlier (on day 0 instead of day 3) to imMKCLs differentiating in shaker flasks led to significantly lower platelet yields (S3 Fig panel G). Taken together, our data uncovered a time window, starting around day 3 of imMKCL differentiation, during which net loss of MT might facilitate platelet biogenesis.

**imMKCLs cultured with VCR show increased platelet production without compromising platelet function in vitro**

In addition to platelet yield studies, we also evaluated the produced platelets using *in vitro* function assays to assess the quality of the iPSC-PLTs. First, we evaluated the agonist-responsiveness by comparing surface-expression of two canonical platelet activation associated antigens, P-Selectin/CD62P and PAC1 (fibrinogen receptor/activated gpIIb/IIIa complex), between control platelets and platelets activated with the platelet agonists ADP and TRAP-6. We observed that platelets made by imMKCLs that had been treated with VCR on day 0 showed a statistically significant and VCR dose-dependent reduction in platelet reactivity compared to platelets made by control imMKCLs, with far fewer agonist-exposed platelets expressing platelet activation-associated antigens (Fig 4A, B). In contrast, ADP/TRAP-6 responsiveness of platelets made from imMKCLs that were treated with either concentration of VCR on day 3 was only marginally reduced in these assays (Fig 4A, B). Similar results were obtained when P-selectin and PAC-1 staining was assessed following stimulation with PMA, another platelet activator (S4 Fig panels A,B). Furthermore, these platelets retained the ability to undergo agonist-stimulated spreading on fibrinogen (S4 Fig panel C).

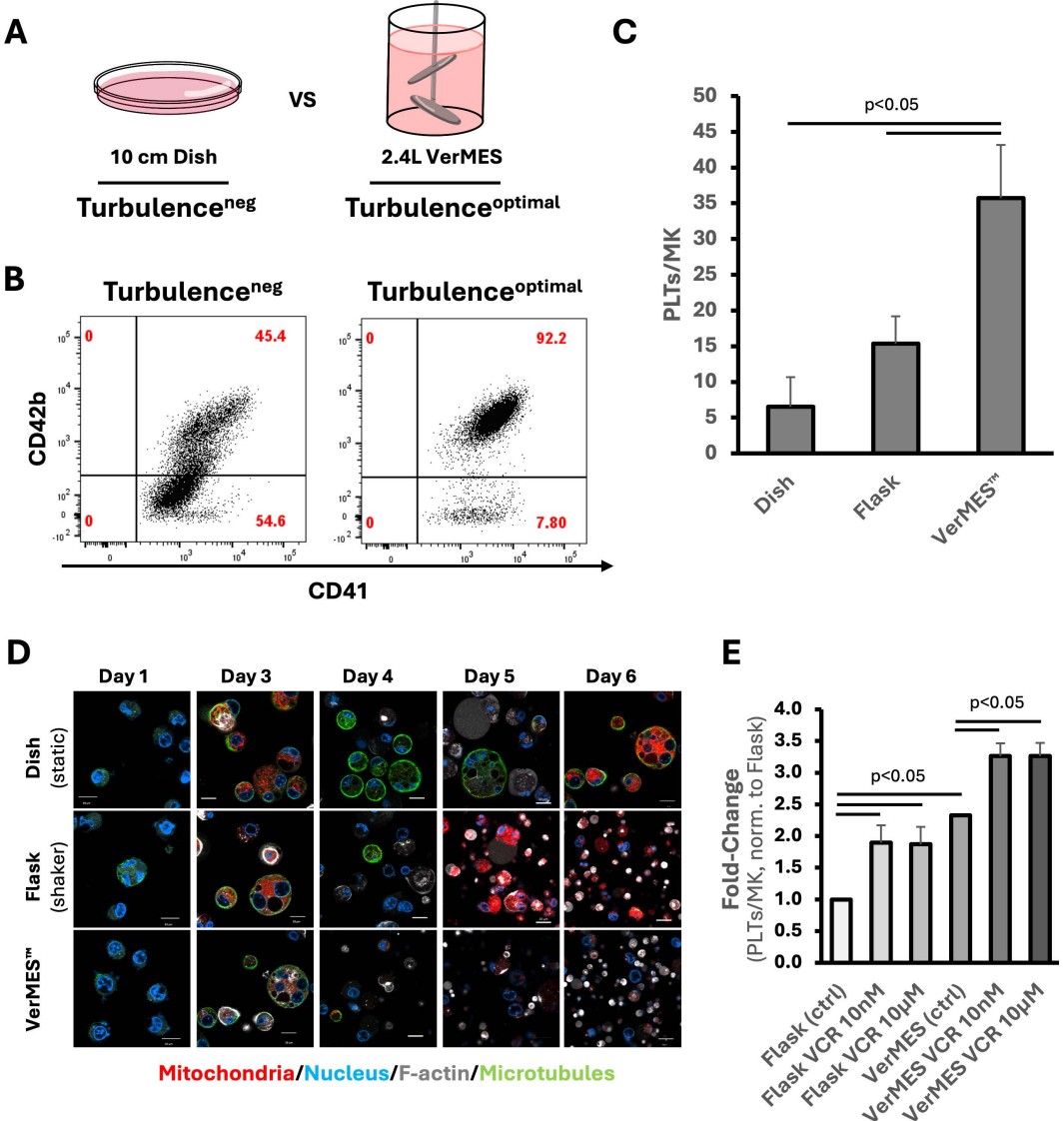

**Fig 3. imMKCL Differentiation under turbulent flow conditions is associated with higher platelet yield and lower megakaryocytic microtubule content.** (A) Overview of imMKCL differentiation/terminal-maturation culture systems: static culture using 10 cm dish is defined as Turbulence[neg] and a 2.4L turbulent-flow-optimized bioreactor (VerMES™) culture is defined as Turbulence[optimal]. (B) Representative dot plots from flow cytometric analyses (gated on particles smaller than imMKCLs) comparing CD41 and CD42b expression profiles between turbulence[neg] (dish, left) and turbulence[optimal] (VerMES™, right). Gating details and data for MK-sized events are provided in SFig3, panel E. (C) Number of platelets generated per imMKCL (CL17) in turbulence[neg] (static dish), turbulence[pos] (shaker flask) and turbulence[optimal] (VerMES™) cultures. Platelets are defined as CD41+/CD42b+ particles. n=3 experiments. Data shows mean (error bars = σ; t-test). (D) Confocal imaging of imMKCLs sampled during maturation on days 1 and 3-6 from static dish (top), shaker flask (middle) and VerMES™ bioreactor cultures revealed a significantly reduced MT signal in cells maturing under turbulent flow providing culture conditions. Red: mitochondria (MitoBright green); blue: nuclei (Hoechst 33342); green: microtubules (SPY650-Tubulin); white: filamentous actin (SPY-555). Scale bar: 10µm (a quantitative analysis is provided in S3 Fig panel F). The paucity of larger imMKCLs after day 3 in shaker-flask and VerMES™ conditions likely is a result of increased proplatelet shedding that has already begun under these conditions. (E) Plot representing the fold-change (normalized to shaker flask control condition) in platelet yields per imMKCLs under different culture- and VCR exposure conditions. Data shows mean (error bars = σ; t-test).

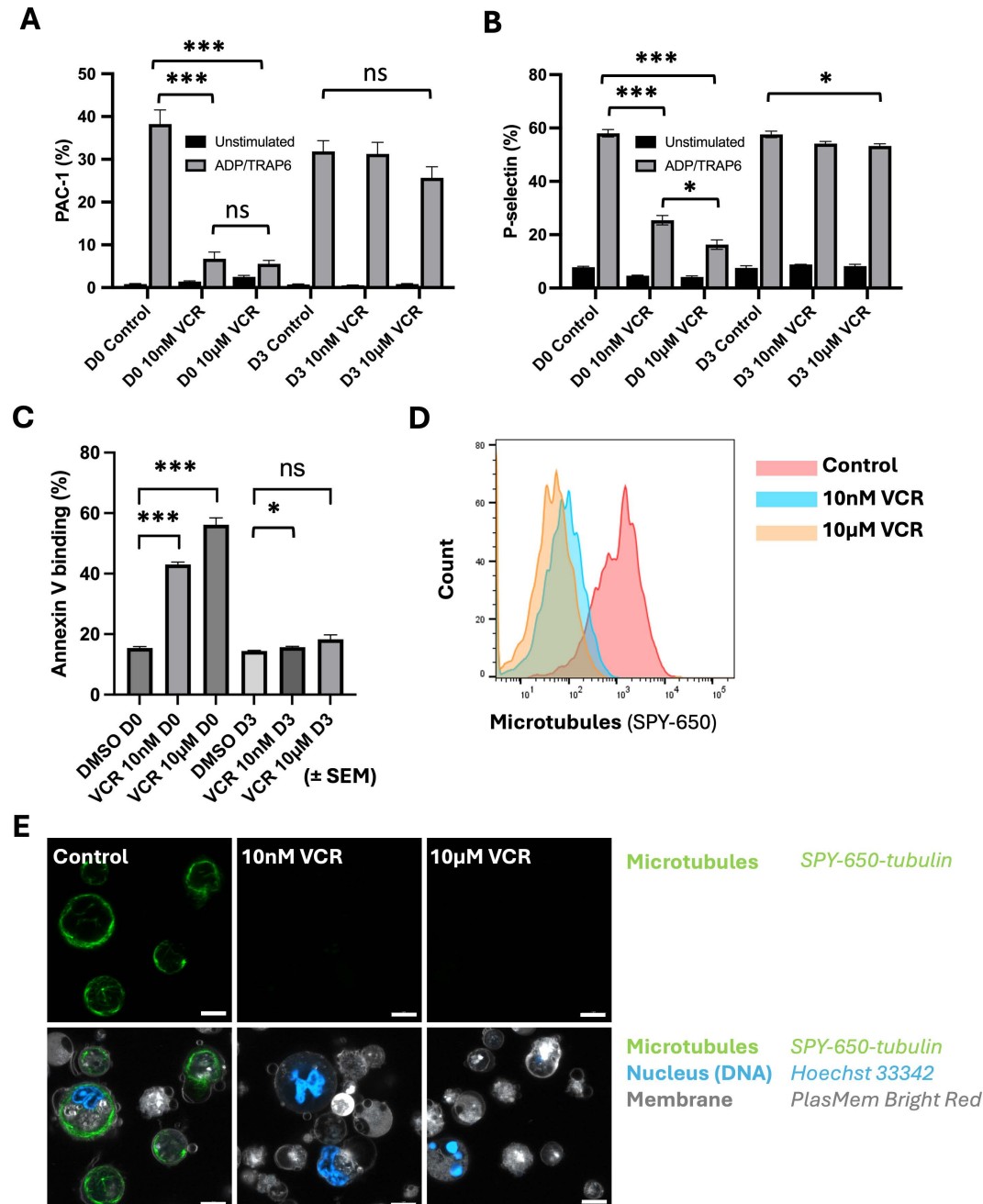

**Fig 4.** imMKCLs exposed to VCR on day 3 of differentiation produce agonist-responsive and Annexin-V-negative platelets. (A) Compiled data of flow cytometric analysis of PAC-1 epitope surface staining of unstimulated and ADP/TRAP-6-stimulated platelets harvested from imMKCLs in shaker-flask cultures, with and without imMKCL exposure to VCR at the indicated doses and times. Data shows means (n = 3; error bars = SEM; t-test: ns p > 0.05; *** p < 0.0001). (B) Compiled data of flow cytometric analysis of CD62P/P-Selectin surface staining of unstimulated and ADP/TRAP-6-stimulated platelets harvested from imMKCLs in shaker-flask cultures, with and without imMKCL exposure to VCR at the indicated doses and times. Data shows means (n=3; error bars = SEM; t-test: * p<0.05, *** p<0.0001). (C) Compiled data for flow cytometry analysis for Annexin V binding on iPSC-PLTs generated in orbital shaking flask upon addition of VCR at concentrations of 10nM and 10 µM on day 0 or day 3. Data shows means (n=3; error bars = SEM; t-test: ns p>0.05; * p<0.05, *** p<0.0001). (D)Representative histogram for day 5 flow cytometric analysis of SPY-650 fluorescence of iPSC-platelets generated in orbital shaking flask upon addition of VCR at concentrations of 10nM and 10 µM on day 3. n=3 experiments. (E) Confocal micrographs (63x) of imMK-CLs on day 6 of differentiation in orbital shaking flask culture. DMSO, 10nM, or 10µM VCR was added to culture on day 3. White pseudocolor=plasma membrane (PlasMem Bright Orange); blue=nuclei(Hoechst 33342); green=microtubules (SPY-650-tubulin). Scale bar: 5µm.

Another distinguishing feature of high-quality resting (non-activated) platelets is paucity of phosphatidylserine in the surface layer of the platelet cell membrane, a marker of apoptosis and premature platelet activation [39]. Therefore, we further examined the quality of iPSC-PLTs using flow cytometric analysis of Annexin V staining. Echoing the results of the platelet agonist studies, we observed that addition of VCR to imMKCL cultures on day 0 resulted in production of lower-quality, Annexin V positive platelets while platelets derived from untreated imMKCLs or those exposed to VCR at either dose on day 3 were largely negative for Annexin V (Fig 4C).

Given that our earlier findings revealed a negative correlation between MT content and platelet yields in our imMKCL system (Fig 3) and considering that vinca alkaloids primarily act by causing a net depolymerization of cellular MTs, we next assessed the impact of VCR treatment on the MT content of imMKCLs. To this end, we stained the cells with SPY-650-tubulin, a cell-permeable and MT-specific dye. Flow cytometric analysis and live-cell imaging confirmed a VCR dose-dependent reduction in MT staining in imMKCLs two and three days after addition of VCR (Fig 4D, E) respectively. Taken together, these results suggest that MT depolymerization through carefully timed and dosed addition of VCR can boost the production of Annexin V-negative iPSC-PLTs that are functional despite a paucity of polymerized MTs.

## VCR-exposed imMKCLs produce hemostatic platelets

Next, we evaluated the ability of day-6 iPSC-PLTs, produced using the VerMES™ bioreactor system in the absence or presence of VCR addition on day 3, to reduce bleeding time in an established *in vivo* hemostasis model [19]. Three batches of iPSC-PLTs were prepared using a 2.4L VerMES™ turbulent-flow bioreactor: without VCR treatment (C=control) or with VCR addition on day 3 at 10 nM (L=low) or at 10 µM (H=high) (Fig 5A). These preparations were then condensed into platelet concentrates of which 4x10^8 iPSC-PLTs were intravenously injected into immune-deficient (NSGS-SGM3) mice rendered thrombocytopenic by gamma-irradiation and treatment with an anti-mouse CD42b (GPIb-alpha) antibody (Fig 5B). Hemostasis (cessation of bleeding within 10 minutes after tail transection) was restored in each group, with no significant differences in bleeding time apparent between the control and the 10 nM VCR cohorts (Fig 5C). We observed a trend towards prolonged bleeding times in mice transfused with the 10 µM VCR iPSC-PLTs. While this effect did not reach statistical significance compared to the control group or the 10 nM group, the 10 µM dose group was the only experimental group that included mice that continued to bleed for over 600 seconds (Fig 5C), in line with our *in vitro* data showing a slightly reduced functionality of iPSC-platelets produced under these conditions (Fig 4B).

To assess the ability of imMKCL-derived platelets to persist in circulation, we quantified human platelet counts using an anti-human CD41+ antibody on peripheral blood samples taken at six time points (30 min to 24 hrs) from the same batch of mice transfused with iPSC-platelets (Fig 5D). We observed that circulation of iPSC platelets from both VCR treatment groups (10 nM and 10 µM) was significantly decreased compared to control iPSC-PLTs at all time points up to 24 hours (Fig 5E, S5 Fig, panel A). Nevertheless, with all groups the counts of iPSC-PLT remained above the threshold for hemostasis in mice [40] for several hours post-infusion. As previously observed [19], the frequency of human platelets increased during the first 2–4 hours, likely due to efficient shedding of platelets *in vivo* from larger structures such as proplatelets.

Another hallmark attributed to normal platelets is the marginal band, a peripheral ring of coiled MTs that is important for maintaining the discoid shape of resting platelets, their ability to respond to activation agonists, and their hemostatic potential [41]. We again used SPY-650-tubulin staining to visualize MTs in iPSC-PLTs before and after injection into mice. Interestingly, while MT staining in hCD41+ platelets was readily detectable in the control for pre-injection samples, the marginal band structure was much reduced in the 10 nM group and virtually non-existent (i.e., below the limit of detection of this assay) in the 10 µM group (S5 Fig pane B, left). Of note, CD41 and MT staining were much reduced in all samples post injection (S5 Fig pane B, right), suggesting that the MTs of these platelets become dynamically rearranged in this *in vivo* model or that CD41^high MT+ iPSC-PLTs get consumed faster than CD41^low MT^low iPSC-PLTs. In summary, exposing imMKCLs to VCR boosts the yield of iPSC-PLTs that, while apparently lacking a prominent MT ring structure and showing

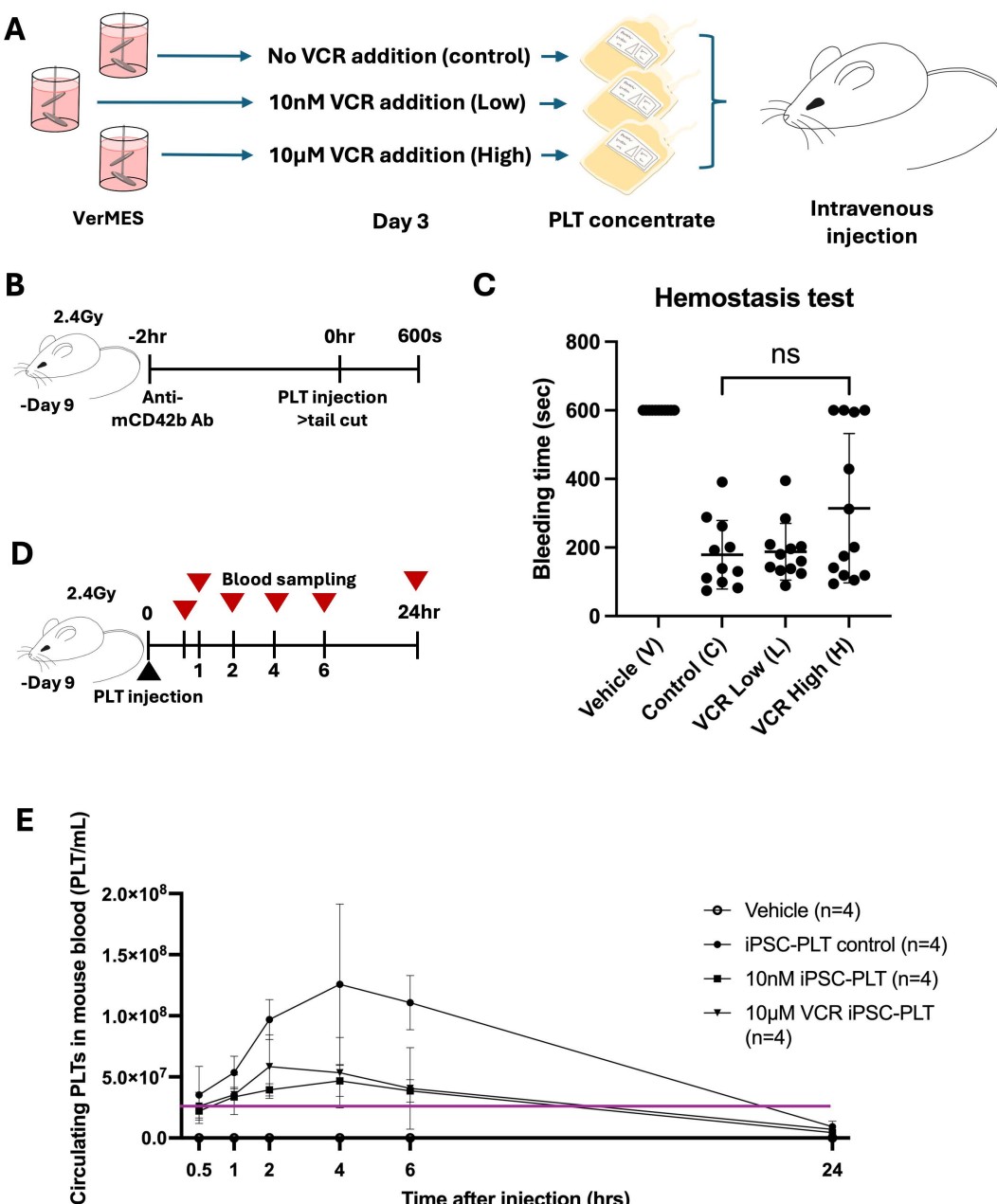

**Fig 5. Platelets produced by imMKCLs exposed to VCR on day 3 of differentiation restore hemostasis in an acute hemorrhage *in vivo* model.**
(A) Schematic diagram of iPSC-PLT production for *in vivo* hemostasis experiments. (B) NSGS-SGM3 mice model for bleeding time experiment. imMKCL derived platelets were intravenously injected into irradiated NSGS-SGM3 mice. 2.4Gy irradiation was performed at -9 days. Anti-mouse CD42b antibody was injected at -2hrs to induce thrombocytopenia. Tail cut was performed immediately after iPSC-PLT injection at position 2.5cm from tail end. Bleeding time was recorded within 600 seconds after the tail incision. (C) Compiled data for bleeding time of mice after tail incision was observed within 600 seconds. Each data point represents a mouse. Groups are vehicle control (V), iPSC-PLT control (C), VCR 10 nM (L), VCR 10 μM (H). Statistical analysis was conducted using Mann-Whitney U Test. Lines represent mean and SD. (D) NSGS-SGM3 mouse model for iPSC-PLT circulation. imMKCL derived platelets were intravenously injected into irradiated NSGS-SGM3 mice. 2.4Gy irradiation was performed at -9 days. Blood sampling was conducted at 0.5, 1, 2, 4, 6, and 24hrs after injection. (E)Representative data for iPSC-PLT circulation in mice. n = 4 mice per group. Each data point represents 4 mice. Groups are vehicle control, iPSC-PLT control, 10 nM VCR iPSC-PLT, 10 μM VCR iPSC-PLT. Purple line represents threshold of circulating platelets to exhibit thrombocytopenic symptoms (error bars = σ).

reduced persistence, are nevertheless competent to respond to agonist stimulation, undergo spreading, and restore hemostasis in an acute bleeding *in vivo* model.

## Discussion

Despite recent advancements in generating platelets from hiPSCs, low yields and high production costs remain formidable obstacles to their clinical translation. In this study, we investigated the variability of platelet production by conditionally immortalized hiPSC-derived megakaryocyte progenitor cells (imMKCLs). Single-cell time-lapse imaging revealed significant heterogeneity in platelet production of individual imMKCLs. While most cells produced a limited number of platelet-sized particles, a minority generated over 50 (and some over 200) (pro)platelets per cell. This variability suggests differences in the maturation and platelet-producing capacity of individual imMKCLs, which may be attributed to stochastic factors influencing terminal differentiation. Of note, platelet production by an imMKCL subclone with increased propensity for platelet biogenesis was also variable at the single cell level, suggesting that factors related to the culture system (e.g., signals or biomechanical forces) rather than genetic or epigenetic drift are chiefly responsible for the observed heterogeneities. The asynchrony and heterogeneity of platelet biogenesis also aligns with previous reports of variable and inefficient *in vitro* MK maturation, polyploidization, and proplatelet formation compared to the *in vivo* situation [15,42], especially when the *in vitro* conditions do not provide differentiation- and proplatelet production-promoting biophysical forces, such as shear and turbulent flow [10,15,43–45]. Understanding the underlying causes of this heterogeneity is crucial for optimizing culture conditions and achieving high platelet yields from each cell.

Recent single-cell RNA sequencing studies have further highlighted the degree of functional heterogeneity within MK/imMKCL populations through the identification of clusters of cells representing distinct phenotypes. These subpopulations include those associated with immature or proliferative states, immune and inflammatory responses, niche interactions, cellular senescence, and platelet biogenesis. [46–49]. The capacity for platelet biogenesis of senescence-biased imMKCLs is diminished and can be restored by inhibiting the senescence effector genes TP53 and CDKN1A/p21$^{CIP1/WAF1}$ [48]. Similarly, heterogeneity of the imMKCL clone 7/Sev2 used in our study is linked to the presence of cells with an immune-MK phenotype characterized by reduced platelet productivity, RALB-mediated elevated secretion of inflammatory cytokines, and downregulation of members of the let-7 family of miRNAs [46]. Intriguingly, platelet biogenesis can be modulated accordingly by employing agonists or inhibitors of the RALB-cytokine axis. These studies therefore provide evidence that addressing the mechanisms that underlie imMKCL heterogeneity can be leveraged to improve platelet yields.

To help address the issue of heterogeneous platelet biogenesis, we conducted a high-content chemical genetics screen for compounds that boost imMKCL proplatelet formation. This screen identified microtubule (MT) destabilizers, particularly vinca alkaloids such as vincristine (VCR) and vinblastine (VBL), as potent enhancers of proplatelet extension formation in this system. The addition of VCR to differentiating imMKCL cultures on day 3 significantly increased platelet yields without compromising *in vitro* functionality across several assays (CD42b expression, responsiveness to activation agonists ADP, TRAP-6, and PMA, ability to spread, and low Annexin V binding). The paucity of surface-exposed phosphatidyl-serine on platelets harvested from VCR-exposed imMKCLs, together with their confirmed *in vitro* functionality, exclude the possibility that VCR treatment or turbulent flow simply induce apoptosis – effects that, if present, could lead to an increase in the number apoptotic bodies and microparticles [50–52] that could otherwise be mistaken for (pro-)platelets due to the small size and presence of platelet markers on such particles. Importantly, the previously established ability of imMKCL-derived platelets to restore hemostasis upon infusion into immune-deficient thrombocytopenic animals [8,19,53] was also fully retained by platelets made from imMKCLs exposed to VCR at the 10nM dose.

While functional, platelets made from VCR-exposed imMKCLs lacked a prominent marginal MT band structure. We cannot completely exclude the possibility that platelets made with VCR may have defects in properties that might be more sensitive to MT perturbations. However, it is worth noting that platelets with a severely diminished marginal MT ring

structure (and consequently, an abnormal spheroid shape) have previously been shown to exhibit a normal *in vivo* lifespan and hemostatic activity [54]. Our results also agree with numerous studies showing that even after exposure to very high concentrations (≥50 µM) of VCR or VBL, platelets remain activatable [55–57] and that the platelets of VCR-treated patients retain the capacity to mediate hemostasis [58,59].

Our data show that timing, in addition to dosing, is critical to avoid adverse effects, with optimal results obtained when VCR was applied on day 3 at a relatively low dose of 10 nM. Interestingly, imMKCLs typically reach their maximal ploidy level by day 3 of terminal differentiation, suggesting that presence of VCR during the preceding endo-mitotic phase (days 0–3) of imMKCL differentiation is detrimental to platelet yields whereas addition to already polyploid imMKCLs on day 3 can facilitate platelet biogenesis. Our results therefore highlight the potential for precisely timed and dosed modulation of MT dynamics to enhance platelet production.

Numerous previous reports observed an increase in platelet counts within a few days following *in vivo* use of MT-destabilizing agents such as vinca alkaloids [30–34] and colcemid [60]. Although the primary target cells and underlying mechanism remain elusive, the observed thrombopoietic effects generally peaked between 3–8 days post-treatment [30,31,33,34,60] and were abrogated by myelo-ablation prior to treatment [34], consistent with our finding that carefully timed and dosed exposure of imMKCLs to VCR *in vitro* can promote terminal differentiation and the formation of functional platelets.

In contrast to these concordant *in vivo* studies, several *in vitro* studies described essential roles for MTs in pro-platelet extension formation and platelet assembly or release [16,61–66]. While this set of reports is seemingly at odds with our results, it is possible that the contradictory accounts are due to differences in the experimental details such as species, media composition, the precise timing or dosing of VCR addition, or unique features intrinsic to the imMKCL system. Nevertheless, they may also reflect our still incomplete understanding of platelet biogenesis and the fact that MKs can produce platelets through multiple alternative mechanisms, such as IL1α-induced stress thrombopoiesis that involves MT disorganization and rupture-type platelet release [67] and platelet biogenesis through MK membrane budding or the release of large protrusions that only contain sparse or disorganized MTs [42,68–70]. Furthermore, efficient proplatelet extension formation has been observed in the marrow of VCR-treated myosin-deficient (Myh9^KO) mice [71]. Since curtailing the activity of ROCK kinases in megakaryocytes appears to promote platelet biogenesis [72] at least in part through suppression of MYH9 [73], the presence of the ROCK inhibitor Y-39983 or Y-27632 in our differentiation medium may have created a cell state that allows the thrombopoietic effect of VCR to manifest.

Our observations of reduced MT-specific staining in imMKCLs cultured under turbulence-inducing conditions further support a link between MT dynamics and platelet production. imMKCL differentiation in systems that generate platelet biogenesis promoting levels of shear stress and turbulence, such as orbital shakers or the VerMES™ bioreactor platform, leads to a decrease in MT fluorescence signals in maturing imMKCLs that correlate with increased platelet yields. Thus, differentiation conditions that properly provide these biophysical forces may mimic the physiological conditions in the bone marrow more closely, where hemodynamic forces contribute to proplatelet elongation and release [42], even in the absence of MTs [71]. The disruption of MTs under turbulent conditions may therefore facilitate the cytoskeletal remodeling necessary for efficient platelet biogenesis. Moreover, the already reduced MT content exhibited by imMKCLs differentiated in shaker flasks or the VerMES™ bioreactor in the absence of VCR treatment may account for the relatively small magnitude of the effect of VCR addition on platelet yields under turbulent flow regimes (40–90% increase, Fig 3E). In contrast, differentiation under static conditions produces imMKCLs with a higher MT content and a much lower propensity for platelet biogenesis at baseline (Fig 2D and Fig 3C), a setting in which VCR has a more dramatic effect (4–7fold increase in proplatelet extension formation and platelet yields; Fig 2D and S1 Fig panel B). Given the high cost of manufacturing hiPSC-derived transfusion products, a relatively modest 40–90% increase in the number of functional hemostatic platelets produced per MK (per volume of culture medium) may enable translation of this platform into a clinically efficacious and economically viable product.

Additional studies will be required to identify – and, if possible, offset – the mechanisms responsible for the somewhat reduced persistence of platelets obtained from VCR-treated imMKCLs, such as by combatting rejection [9] or cell death [74]. Nevertheless, even if their lifespan in circulation cannot be augmented, the proven hemostatic activity of these platelets suggests therapeutic utility in applications such as treatment of acute hemorrhage.

Taken together, our study underscores the importance of MT dynamics in megakaryocyte maturation and platelet production. By leveraging chemical modulation with VCR, we provide a simple method to boost the yield of functional platelets from hiPSC-derived imMKCLs with an FDA-approved compound. Our findings provide novel insights for the development of scalable platelet manufacturing processes for transfusion therapies. Additional research is warranted to further optimize specific aspects of VCR-augmented platelet production systems, including platelet yield, functionality, and persistence. Future studies could also explore strategies to balance temporary destabilization of megakaryocytic MTs for enhanced production with the benefit of preserving MT structures within platelets. For example, genetic circuits, such as eToeholds or DART-VADER [75,76] that work orthogonally to the DOX system and that can be designed to become activated precisely when a differentiating cell reaches a predetermined stage of maturation (trigged by onset of expression of specific endogenous RNAs) could be employed to induce reversible MT depolymerization.

## Supporting information

**S1 Fig. High-throughput chemical genetics screen and titration of hit compounds vinblastine and vincristine.** (A) Histogram of the size distribution of imMKCLs (related to Fig 1B; diameters estimated from 20x confocal timecourse imaging data with membrane-dye stained imMKCLs). (B) Rank-order plot comparing platelet productivity between individual imMKCLs from a subclone with increased platelet productivity (subclone 38; red line) and imMKCLs from the bulk population (CL17, green line). Time-course imaging was performed as before (Fig 1B) except that confocal imaging was carried out with as 4x lens (due to the lower spatial resolution and fluorescence signal sensitivity under these experimental conditions, the number of PLPs detected is proportional to but likely lower than the actual number of PLPs produced by each cell). While cells from the subclone produce more PLPs (p = 3.449E-13; Student's t-test), they still include cells the produce few if any PLPs. Representative screenshots (frame # 267) from the bulk and the subclone time-course datasets are shown next to the plot. A video with the time-course imaging data is included in the Supplemental Data section. (C) Stains/dyes used in the high-content chemical genetics screen (images from a spreading phenotype hit well). (D) Frequency distribution of compounds in the screening library. The screening library included several commercial compound libraries, resulting in some compounds being represented multiple times. (E) Image representing processing and analysis steps used to classify pixels as belonging to individual imMKCLs (including pro-platelet extensions): background-corrected, cropped, and pseudo-colored RGB image representing the central region of a vinca alkaloid hit well (left), cell mask created to capture imMKCLs and their pro-platelet-like extensions but excluding debris and detached platelet-like particles (middle), and cells after removal of the debris/platelet-like particles and empty areas (right). Scale bar = 100µm. (F) Fiji and Python scripts were used to process, analyze, and plot the high-content high-throughput confocal imaging data. MKs and PLPs appear as distinct populations in a 2D scatter plot of the membrane content over DNA (left-most panel). Polyploidization of terminally differentiating imMKCLs is apparent (2nd panel). The membrane content (average PKH26 intensity per pixel; 3rd panel) and cell size (number of PKH26-positive pixels per cell; last panel) of imMKCLs correlates in the expected manner with their DNA content (differentiation state). All axes except for the ploidy histogram Y axis are on $\log_{10}$-scale.
(TIFF)

**S2 Fig. Hit validation, titration, and effect of delayed application.** (A) Representative images (20x confocal) showing PKH426/Hoechst stained imMKCLs (D6) after treatment with DMSO or hit compound (VCR) under static conditions. Arrows mark imMKCLs with proplatelet extensions. (B) Titration of hit compounds vincristine (VCR) and vinblastine (VBL)

under static culture conditions (384w plate). The fold-change (over control) of the number of platelet-like particles per cell is plotted over compound concentration. Processing and quantification of the 20x confocal images were carried out using a FIJI script. (C) Z-score plots of the data shown in (B). With both compounds, Z-scores of ≥6 were obtained with concentrations as low as 6.1nM. (D) Representative images (10x brightfield) of day 6 proplatelet formation in static imMKCL differentiation culture. 10nM VCR was added on day 3 or day 5. Scale bars represent 200µm. Black-framed views of 2x zoomed-in sections are included for better visibility.
(TIFF)

**S3 Fig. Effect of VCR use on MK ploidy and size.** (A) Representative flowcytometry histograms showing day 4 maturing imMKCL ploidy under orbital shaking flask culture condition. DMSO, 10nM VCR, or 10µM VCR was added on day 3. Gates show 2N, 4N, 8N, >16N. (B) Bar plot shows compiled data of day 4 maturing imMKCL under orbital shaking flask culture condition. DMSO, 10nM VCR, or 10µM VCR was added on day 3. Experiments were conducted n = 3 times. Error bars represent SEM and statistical analysis was conducted using one way-Anova test of variance. (C) Representative flow cytometry density plots for imMKCL cell size on day 4 of differentiation under orbital shaking flask culture condition (FSC-A). Gates show <100K (left, small), 100K-150K (middle, medium), >150K (right, large). Either DMSO, 10nM VCR, or 10µM VCR were added on day 3. (D) (E) Bar plot shows compiled data of day 4 differentiating imMKCLs under orbital shaking flask culture condition. DMSO, 10nM VCR, or 10µM VCR were added on day 3. Experiments were conducted n = 3. Error bars represent SEM and statistical analysis was conducted using one way-Anova test of variance. *$p < 0.05$, *** $p < 0.001$, ****$p < 0.0001$. Gating Strategy and Overlay of PLTs (pro)platelets) and MKs (imMKCLs) sampled on day 6 from dish (turbulence⁻) and VerMES™ (turbulence⁺) differentiation cultures (related to Fig 3B, C). (F) FIJI/ImageJ-based quantification of confocal imaging signal representing the MT content of imMKCLs sampled from day 4 static-dish and VerMES™ differentiation cultures (related to Fig 3D; Student's t-Test, ***$p < 0.001$). (G) Effect on platelet-yields of early (day 0) addition of VCR to imMKCLs differentiating in shaker flasks. Platelet yields are graphed as fold-change in yield (PLTs/MK) over control (D0, DMSO). Data shows means (n = 3; error bars = σ; p-values: t-test).
(TIFF)

**S4 Fig. Effect of VCR use on iPSC-PLT stimulation with PMA and agonist-induced spreading.** (A) Compiled data of flow cytometric analysis of PAC-1 epitope surface staining of unstimulated and PMA stimulated platelets harvested from imMKCLs in shaker-flask cultures, with and without imMKCL exposure to VCR at the indicated doses and times. Data shows means (n = 3; error bars = SEM; t-test: ns $p > 0.05$; *** $p < 0.0001$). (B) Compiled data of flow cytometric analysis of CD62P/P-Selectin epitope surface staining of unstimulated and ADP/TRAP-6 stimulated platelets harvested from imMKCLs in shaker-flask cultures, with and without imMKCL exposure to VCR at the indicated doses and times. Data shows means (n = 3; error bars = SEM; t-test: * $p < 0.05$; ** $p < 0.001$; *** $p < 0.0001$). (C) Confocal micrographs of platelet spreading. Human PLTs, iPSC-PLTs, 10nM VCR iPSC-PLTs, or 10µM VCR iPSC-PLTs were seeded on human fibrinogen coated dishes with or without ADP/Trap6. Red: hCD41 (anti-hCD41-APC); white: filamentous actin (SPY-555). Scale bar: 10µm.
(TIFF)

**S5 Fig. Persistence *in vivo* and marginal band analysis.** (A) iPSC-PLT circulation in mouse blood at time points: 10min, 30min, 1hr, 2hrs, 4hrs, 6hrs, 24hrs. n = 3 mice per group. Error bars represent SD. Related to Fig 5E. (B) Confocal micrographs of iPSC-PLTs from *in vivo* circulation experiment (performed in NSGS-SGM3 mice). Pre-injection(left) and post injection (right). Staining shows microtubules with SPY650-Tubulin (red) and with anti-hCD41 antibody (green). Scale bars show 10µm. Purple arrows point to prominent marginal band MTs present in control iPSC-PLTs. The tubulin staining signal is also shown in log-scale (lower half) to better visualize presence of residual MT staining and marginal band structures in 10nM VCR iPSC-PLTs prior to injection (blue arrows).
(TIFF)

**S1 Table.  Compound Libraries.** Summary of the compound libraries used in the high throughput screen.
(PDF)

**S2 Table.  Reagents.** Vendor and catalog number or compound ID code for staining reagents, antibodies, and small molecules and other chemicals.
(PDF)

**S1 File.  Single cell platelet production timelapse movie.** Timelapse movie showing a single, membrane-dye stained imMKCL (from dataset shown in Figure 1B) that produces hundreds of platelet-like particles in a burst-like fashion. 20x confocal imaging.
(AVI)

**S2 File.  Single cell platelet production timelapse movie: bulk and subclone.** 4x confocal imaging timelapse datasets showing multiple imMKCL from the original imMKCL line (bulk culture, left) and from subclone of that line. A rank order plot of the per-imMKCL platelet productivity rates is shown in S1 Fig. panel B.
(AVI)

**S3 File.  Figures summary, spreadsheets, and names of scripts.**
(XLSX)

**S4 File.  FCS files for Fig 3B/ S3 Fig E.**
(ZIP)

**S5 File.  FCS files for Fig. 4D.**
(ZIP)

**S6 File.  FCS files for S3 Fig. A, B.**
(ZIP)

**S7 File.  FCS files for S3 Fig. C, D.**
(ZIP)

**S8 File.  Scripts.**
(ZIP)

## Acknowledgments

The authors express their gratitude to Dr. Ronald Mathieu and the Boston Children's Hospital flow cytometry core facility for excellent flow cytometry service and to the members of the George Q. Daley, Leonard I. Zon, Trista North, Andrew L. Frelinger, Martha Sola-Visner, Joseph Italiano, Bruce Furie, and Thorsten M. Schlaeger laboratories, including Arunoday Bhan, for valuable suggestions and discussions as well as technical assistance and Fig 1A artwork.

## Author contributions

**Conceptualization:** Koji Eto, Thorsten M. Schlaeger.

**Data curation:** Thorsten M. Schlaeger.

**Formal analysis:** Thorsten M. Schlaeger.

**Funding acquisition:** Leonard I. Zon, George Q. Daley, Koji Eto, Thorsten M. Schlaeger.

**Investigation:** Emiri Nakamura, Yasuo Harada, Trevor Bingham, Christian Skorik, Anjali Jha, John Atwater, Natsumi Higashi, Kosuke Fujio, Mariko Ishiguro, Haruki Okamoto, Andrew L. Frelinger, Koji Eto, Thorsten M. Schlaeger.

**Methodology:** Christian Skorik, Andrew L. Frelinger, Thorsten M. Schlaeger.

**Project administration:** Koji Eto, Thorsten M. Schlaeger.

**Resources:** Andrew L. Frelinger, Koji Eto, Thorsten M. Schlaeger.

**Software:** Thorsten M. Schlaeger.

**Supervision:** Leonard I. Zon, George Q. Daley, Koji Eto, Thorsten M. Schlaeger.

**Validation:** Trevor Bingham, John Atwater, Andrew L. Frelinger, Thorsten M. Schlaeger.

**Visualization:** Emiri Nakamura, Thorsten M. Schlaeger.

**Writing – original draft:** Emiri Nakamura, Thorsten M. Schlaeger.

**Writing – review & editing:** Emiri Nakamura, Andrew L. Frelinger, Koji Eto, Thorsten M. Schlaeger.

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
