## [Decision Letter · Decision Letter 0]

Dear Dr. Schlaeger,

We look forward to receiving your revised manuscript.

Kind regards,

Francesco Bertolini, MD, PhD

Academic Editor

PLOS ONE

Journal Requirements:

2.  Please update your submission to use the PLOS LaTeX template. The template and more information on our requirements for LaTeX submissions can be found at http://journals.plos.org/plosone/s/latex .

3. Thank you for submitting your work to PLOS ONE. We note that you have not mentioned the source imMKCL cell lines used in this study. Before we can continue with your submission, kindly mention the details in the Method section. Thank you for your attention to this request. We look forward to hearing from you.

“This work was made possible by the generous support of the Boston Children’s Hospital Stem Cell Program, the NIDDK (U54DK110805) and Megakaryon Inc. This work was supported in part by the Core research center for next-generation medicine utilizing Cell and gene therapy grant (23bm1323001h0001 to K.E.) from AMED, the CiRA Foundation Fund (to K.E.), a grant-in-aid for scientific research (KIBAN S, 21H05047, to K.E) from the Japan Society for the Promotion of Science (JSPS), and collaborating project funding from Megakaryon Co. Ltd. (to K.E., A. F., and T. S.).”

“The authors express their gratitude to Dr. Ronald Mathieu and the Boston Children’s Hospital flow cytometry core facility for excellent flow cytometry service and to the members of the George Q. Daley, Leonard I. Zon, Trista North, Andrew L. Frelinger, Martha Sola-Visner, Joseph Italiano, Bruce Furie, and Thorsten M. Schlaeger laboratories, including Arunoday Bhan, for valuable suggestions and  discussions as well as technical assistance. This work was made possible by the generous support of the Boston Children’s Hospital Stem Cell Program, the NIDDK (U54DK110805) and Megakaryon Inc. This work was supported in part by the Core research center for next-generation medicine utilizing Cell and gene therapy grant (23bm1323001h0001 to K.E.) from AMED, the CiRA Foundation Fund (to K.E.), a grant[1]136 in-aid for scientific research (KIBAN S, 21H05047, to K.E) from the Japan Society for the Promotion of Science (JSPS), and collaborating project funding from Megakaryon Co. Ltd. (to K.E., A. F., and T. S.).”

“This work was made possible by the generous support of the Boston Children’s Hospital Stem Cell Program, the NIDDK (U54DK110805) and Megakaryon Inc. This work was supported in part by the Core research center for next-generation medicine utilizing Cell and gene therapy grant (23bm1323001h0001 to K.E.) from AMED, the CiRA Foundation Fund (to K.E.), a grant-in-aid for scientific research (KIBAN S, 21H05047, to K.E) from the Japan Society for the Promotion of Science (JSPS), and collaborating project funding from Megakaryon Co. Ltd. (to K.E., A. F., and T. S.).”

6. Thank you for stating the following in the Competing Interests section:

“K.E. is a founder of Megakaryon and a member of its scientific advisory board without salary and receives research funding from Megakaryon, Otsuka Pharmaceutical, and Kyoto Manufacturing Co. T.S. received research funding from Megakaryon. Y.H. and K.F. are currently employed by Otsuka Pharmaceutical CO., Ltd., Osaka, Japan. The remaining authors declare no competing financial interests”

7. In the online submission form, you indicated that “All relevant data are within the manuscript and its Supporting Information files. Image processing scripts and raw imaging data from the chemical genetic screen and time-course imaging experiments are available upon request (corresponding author)”

8. We note that you have included the phrase “data not shown” in your manuscript. Unfortunately, this does not meet our data sharing requirements. PLOS does not permit references to inaccessible data. We require that authors provide all relevant data within the paper, Supporting Information files, or in an acceptable, public repository. Please add a citation to support this phrase or upload the data that corresponds with these findings to a stable repository (such as Figshare or Dryad) and provide and URLs, DOIs, or accession numbers that may be used to access these data. Or, if the data are not a core part of the research being presented in your study, we ask that you remove the phrase that refers to these data.

9. We notice that your supplementary figures are uploaded with the file type 'Figure'. Please amend the file type to 'Supporting Information'. Please ensure that each Supporting Information file has a legend listed in the manuscript after the references list.

10. We notice that your supplementary tables are included in the manuscript file. Please remove them and upload them with the file type 'Supporting Information'. Please ensure that each Supporting Information file has a legend listed in the manuscript after the references list.

Reviewers' comments:

Reviewer's Responses to Questions

**Comments to the Author**

1. Is the manuscript technically sound, and do the data support the conclusions?

Reviewer #1: Yes

Reviewer #2: Yes

2. Has the statistical analysis been performed appropriately and rigorously?

Reviewer #1: No

Reviewer #2: Yes

3. Have the authors made all data underlying the findings in their manuscript fully available?

Reviewer #1: Yes

Reviewer #2: Yes

4. Is the manuscript presented in an intelligible fashion and written in standard English?

Reviewer #1: Yes

Reviewer #2: No

Reviewer #1: The study by Nakamura et al. investigated the maturation and platelet generation capacity of immortalized megakaryocyte-like cells (imMKCLs) using single-cell time-course imaging and a chemical genetics screen. The authors identified microtubule-destabilizing agents, such as vincristine (VCR), as enhancers of proplatelet formation, which reduced microtubule content in megakaryocytes, similar to turbulence. These findings suggest that reduction in microtubules could be a central trigger for proplatelet formation. While the VCR-treated platelets lacked the characteristic marginal band, they remained functional, as demonstrated by their response to agonists and ability to restore haemostasis in an in vivo thrombocytopenia model.

The manuscript presents an original, interesting and well-performed study, but there are some inconsistencies, missing quantifications, and areas requiring clarification.

The results describe the maturation process as "fast," yet the methods indicate a six-day maturation period (D0-D6). It would be helpful to use more precise language to describe the maturation timeline. Do the authors observe terminally mature cells already at D1? Additionally, it is unclear when imaging was initiated during this process, whether it began at the start of maturation (D0), after some days of differentiation, or exclusively on D6. It is also unclear whether shaking conditions were maintained during the 45-hour imaging period, what speed of shaking was applied, and whether between D0-D6 the cells were further diluted to maintain a specific cell density or if the media was replaced at any point during the differentiation process. Clarifying these conditions would be important for interpreting the asynchronous platelet production observed. Furthermore, there is an inconsistency between the stated dimensions of the microwells in the figure legend (200x200x100 μm) and the methods section (200x200x200 μm).

In addition, the authors should provide quantification, which is likely already available to them: What percentage of cells released less than 10 or over 50 PLPs? How does the size of the cells correlate with PLP formation activity, do larger cells release more PLPs, or is there no correlation? Additionally, how homogeneous are the cell diameters in these experiments? The authors should provide details on the percentage of cells that fall into size categories such as smaller than 15 μm, between 15-25 μm, and 25-35 μm or larger. While the authors state in the Fig. 1 legend that both PLP formation activity and MK maturation are heterogeneous, they do not provide quantitative data supporting the second claim. Including these analyses would provide valuable insight into the observed variability.

The results in Fig. 2D are indeed impressive, however the statement “these results demonstrate the effectiveness of our chemical genetics screening approach in identifying compounds … that affect imMKCL terminal differentiation” may be overstated. While the results suggest that these compounds increase PPF, it is unclear if they also enhance megakaryocyte differentiation. Ploidy or cell size are not explicitly reported and should be assessed to substantiate claims. It would be also valuable to test whether the observed effects are specific to prolonged treatment or if adding the compounds only during the last 24 hours yields similar outcomes. Statistical analysis for Fig. 2D is missing and should be included. Could the authors repeat the single-cell time course confocal imaging studies (Fig. 1B) to determine whether VCR increases the percentage of cells generating proplatelets or even enhance PLP production beyond the levels of the highest-producing control cells? Additionally, the Fig. S1B legend contains an incomplete sentence, and the concentration unit is missing in S1E.

In Fig. 3D, the images show smaller cells under turbulence conditions, with increased numbers of naked nuclei and F-actin-positive structures. Could the authors clarify why cells are smaller at earlier time points under turbulence conditions? If turbulence enhances maturation, one might expect larger cells. The authors should discuss whether the observed changes reflect physical fragmentation or active maturation, as the flow cytometry plot suggests the latter, and speculate on the potential mechanism driving this process. Is Fig. 3B gated on the larger cells, or does it represent the signal from all cells? It would be beneficial to duplicate this plot separately for MKs and PLPs to clarify the contributions of each population. Additionally, could the authors provide quantification for Fig. 3D to support the visual observations?

Both turbulence (Fig. 3D) and VCR (Fig. 4E) appear to reduce microtubule levels, which could suggest that combining these conditions might not lead to further improvements in platelet yield. Did the authors test this scenario?

The authors should directly acknowledge the complete lack of the marginal band in VCR-treated platelets when describing Fig. S3B. This is a critical observation that should be transparently reported, as it has significant implications for platelet function, particularly in processes dependent on the marginal band. The authors could also consider testing their platelets in assays where microtubules play a central role, such as spreading, shape change, and contractility, to further support their claim that platelets generated with VCR show increased production without compromising function. Alternatively, they could refine their claims by clarifying that while the platelets performed comparably to controls in the assays tested, not all relevant functions were evaluated, and highlight in the discussion that microtubule-dependent processes could potentially be affected by VCR treatment.

The nature of the statistical test is only described for Fig. 5C. The authors should clarify the specific statistical tests used for each subpanel where statistics were applied to ensure transparency and reproducibility.

Reviewer #2: Enhancing the effectiveness of producing platelets from megakaryocytes (MKs) in vitro presents a major challenge and is of great importance. In this manuscript, Nakamura et al. performed high-throughput screening and identified microtubule (MT) destabilizing agents, including the vinca alkaloid vinblastine and vincristine (VCR), as potential promoters of imMKCL maturation and proplatelet formation. The overall study is interesting and significant for improving platelet yield from MKs. However, additional evidence is necessary to substantiate the claims made by the authors.

Major concerns

1. The authors found that imMKCLs sampled from culture systems with turbulent flow (turbulence+) showed reduced microtubule staining in terminally maturing imMKCLs compared to those sampled from static culture conditions (turbulence-), especially after day 3 (Fig 3D), similar to the expected phenotype of administering VCR. And the platelet yield per MK from the turbulence+ group is much higher than turbulence- group. How does the platelet yield per MK compare between the VCR treatment and the turbulence+ group? Can chemical compounds effectively replace the complex bioreactor devices?

2. The authors compared surface-expression of P-Selectin/CD62P and PAC1 between control platelets and platelets activated with the platelet agonists ADP and TRAP-6. Since platelet are easily activated during the culturing and experimental processing, please provide the data for P-Selectin/CD62P and PAC1 expression without stimulation by any agonist.

3. After transfused into mice, the circulation of VCR-iPSC-PLTs from both VCR treatment groups was significantly decreased compared to control iPSC-PLTs at all time points up to 24 hours. What is the possible mechanism underlying this phenomenon?

4. Although exposing imMKCLs to VCR boosts the yield of iPSC-PLTs, but VCR-iPSC-PLTs show reduced persistence and perform not as well as control iPSC-PLTs, as shown in Fig 5E and S3A. What are the advantages of using VCR in platelet-producing culture systems with turbulent flow?

Minor concerns

1. The culture condition of the turbulence+ group in Fig 3C is similar to the DMSO group in Fig 4A, but the platelet yield per MK of the DMSO group is much lower than that of the turbulence+ group. Does DMSO play a role in decreasing platelet yield?

2. Some language inaccuracies are present in the text. Please refine the manuscript.

**Do you want your identity to be public for this peer review?** For information about this choice, including consent withdrawal, please see our Privacy Policy

Reviewer #1: No

Reviewer #2: No

---

## [Author Response · Author response to Decision Letter 1]

18 Apr 2025

(please see PDF version for response with revised manuscript text included as images)

AUTHOR RESPONSE

We would like to thank the editorial team and the reviewers for their careful analysis and constructive feedback. We have prepared a substantially revised manuscript with additional data in response to the many excellent points and suggestions provided.

Reviewer 1:

The study by Nakamura et al. investigated the maturation and platelet generation capacity of immortalized megakaryocyte-like cells (imMKCLs) using single-cell time-course imaging and a chemical genetics screen. The authors identified microtubule-destabilizing agents, such as vincristine (VCR), as enhancers of proplatelet formation, which reduced microtubule content in megakaryocytes, similar to turbulence. These findings suggest that reduction in microtubules could be a central trigger for proplatelet formation. While the VCR-treated platelets lacked the characteristic marginal band, they remained functional, as demonstrated by their response to agonists and ability to restore haemostasis in an in vivo thrombocytopenia model.

The manuscript presents an original, interesting and well-performed study, but there are some inconsistencies, missing quantifications, and areas requiring clarification.

The results describe the maturation process as "fast," yet the methods indicate a six-day maturation period (D0-D6). It would be helpful to use more precise language to describe the maturation timeline.

Response: Indeed, our description of the process as ‘fast’ was somewhat ambiguous. We have modified this paragraph using more precise language to clearly communicate that this protocol is faster than protocols that require each hiPSC production run to start from hiPSCs:

Do the authors observe terminally mature cells already at D1?

Response: No, on day 1 (D1), the cells are not yet fully mature. We have added clarifying language to the introduction:

Additionally, it is unclear when imaging was initiated during this process, whether it began at the start of maturation (D0), after some days of differentiation, or exclusively on D6.

It is also unclear whether shaking conditions were maintained during the 45-hour imaging period, what speed of shaking was applied, and whether between D0-D6 the cells were further diluted to maintain a specific cell density or if the media was replaced at any point during the differentiation process. Clarifying these conditions would be important for interpreting the asynchronous platelet production observed. Furthermore, there is an inconsistency between the stated dimensions of the microwells in the figure legend (200x200x100 μm) and the methods section (200x200x200 μm).

Response: Timecourse imaging experimental procedures are described in more detail in the methods section and figure legend to address these points and the incorrect dimension of the microwells was corrected in the method section. In brief: timecourse imaging was conducted with plates containing 200x200x100µm (W*D*H) microwells, under static conditions, starting with pre-differentiated cells (~ day 3.5) that no longer divide (see above) – i.e., no dilution was required to maintain a density of 1 imMKCL per microwell.

In addition, the authors should provide quantification, which is likely already available to them: What percentage of cells released less than 10 or over 50 PLPs? How does the size of the cells correlate with PLP formation activity, do larger cells release more PLPs, or is there no correlation? Additionally, how homogeneous are the cell diameters in these experiments? The authors should provide details on the percentage of cells that fall into size categories such as smaller than 15 μm, between 15-25 μm, and 25-35 μm or larger. While the authors state in the Fig. 1 legend that both PLP formation activity and MK maturation are heterogeneous, they do not provide quantitative data supporting the second claim. Including these analyses would provide valuable insight into the observed variability.

Response: We have re-analyzed the dataset to address these important questions. The new data are presented and discussed in the Results section of the revised manuscript and a size distribution histogram was added to the supplemental dataset as S1 Fig A.

The results in Fig. 2D are indeed impressive, however the statement “these results demonstrate the effectiveness of our chemical genetics screening approach in identifying compounds … that affect imMKCL terminal differentiation” may be overstated. While the results suggest that these compounds increase PPF, it is unclear if they also enhance megakaryocyte differentiation.

Ploidy or cell size are not explicitly reported and should be assessed to substantiate claims.

Response: The results shown in Figure 2D represent endpoint (day 6) data. At this timepoint, the cells have already undergone significant differentiation- and proplatelet-biogenesis associated shape changes, which makes it difficult to extract meaningful cell size data from this dataset. We therefore performed new flow-cytometric analyses to estimate the ploidy and cell size (S3 Fig). In brief: VCR-exposed imMKCLs do not appear to be more mature, at least not as determined by these paramters. We have removed statements that suggested a maturity-boosting effect and added the following statement for clarification:

It would be also valuable to test whether the observed effects are specific to prolonged treatment or if adding the compounds only during the last 24 hours yields similar outcomes

Response: The revised manuscript includes new data showing that addition of VCR on day 5 (last 24h) does not boost platelet biogenesis:

Statistical analysis for Fig. 2D is missing and should be included.

Response: The missing statistical method (Student’s t-test) and p-values (<1E-5) are now included in the revised manuscript.

Could the authors repeat the single-cell time course confocal imaging studies (Fig. 1B) to determine whether VCR increases the percentage of cells generating proplatelets or even enhance PLP production beyond the levels of the highest-producing control cells?

Response: Figure 2D already addresses the first point: the percentage of cells generating proplatelets is indeed increased after exposure of the cells to vinca alkaloids (VCR, vinblastine). The revised manuscript also includes larger fields of view of this dataset (S2 Fig panel A) to display a wider range of representative cells that have undergone differentiation under control or VCR conditions. This data also shows that, on average, these extensions tend to be larger and more elaborate compared to those produced by control cells.

Additionally, the Fig. S1B legend contains an incomplete sentence, and the concentration unit is missing in S1E.

Response: These inadvertent omissions have been corrected in the revised manuscript.

In Fig. 3D, the images show smaller cells under turbulence conditions, with increased numbers of naked nuclei and F-actin-positive structures. Could the authors clarify why cells are smaller at earlier time points under turbulence conditions? If turbulence enhances maturation, one might expect larger cells.

Response: We have observed that differentiating imMKCLs start producing iPSC-PLTs as early as day 3.5-4. The cells being smaller could indeed be due to imMKCLs having already shed more proplatelets (DNA-low F-actin positive structures) at earlier time points in turbulence-positive cultures.

The authors should discuss whether the observed changes reflect physical fragmentation or active maturation, as the flow cytometry plot suggests the latter, and speculate on the potential mechanism driving this process. Is Fig. 3B gated on the larger cells, or does it represent the signal from all cells? It would be beneficial to duplicate this plot separately for MKs and PLPs to clarify the contributions of each population. Additionally, could the authors provide quantification for Fig. 3D to support the visual observations?

Response:

The observed changes are likely to reflect increased normal platelet biogenesis involving shedding of platelet-sized particles as well as larger fragments that have the potential to become fragmented into platelet-sized particles. Physcal forces can promote proplatelet extension formation, shedding, and fragmentation. However, excessive physical forces and other processed such as cell death can also result in increased particle formation. We show that the produced particles have many characteristics of naïve (not yet activated), reactive (activatable), and Annexin-V-negative (pro)platelets. Taken together, these finding indicate that the majority of the produced particles are not debris or microparticles. Further supporting this interpretation, the produced particles have the capacity to undergo agonist-induced spreading and to restore hemostasis in vivo.

Our new supplementary MK size data (S3 Fig A-D) also supports this observation as increased shedding is likely to result in reduced MK size.

Fig.3B is gated on the entire PLT-sized population and excludes MK sized populations. The complete gating strategy for Fig. 3B as well as results for MK-sized events are provided in S3 Fig. E. and quantification for Fig 3D Day 4 MT MFI have been added as a new supplemental figure panels,

Both turbulence (Fig. 3D) and VCR (Fig. 4E) appear to reduce microtubule levels, which could suggest that combining these conditions might not lead to further improvements in platelet yield. Did the authors test this scenario?

Response: Indeed, our data show that this is the case: the effect size (fold-increase in platelet yields) is lower when VCR is used in the context of turbulent flow compared to static conditions. The revised manuscript includes a fold-change plot (Fig 3E) to more clearly show this. We have expanded the Discussion section to to provide a more detailed explanation of these findings and clarify their interpretation:

The authors should directly acknowledge the complete lack of the marginal band in VCR-treated platelets when describing Fig. S3B. This is a critical observation that should be transparently reported, as it has significant implications for platelet function, particularly in processes dependent on the marginal band. The authors could also consider testing their platelets in assays where microtubules play a central role, such as spreading, shape change, and contractility, to further support their claim that platelets generated with VCR show increased production without compromising function. Alternatively, they could refine their claims by clarifying that while the platelets performed comparably to controls in the assays tested, not all relevant functions were evaluated, and highlight in the discussion that microtubule-dependent processes could potentially be affected by VCR treatment.

Response: The marginal MT band structure is not necessarily completely absent in platelets derived from VCR-exposed imMKCLs. The revised manuscript includes log-transformed images to make the presence of these structures in some platelets easier to see. The observed persistence of marginal MT bands in platelet derived from imMKCLs exposed to low (10nM) concentrations of VCR is consistent with reports of persisting marginal MT bands in platelets treated with high (10µM) concentrations of VCR (see PMID 25701419; 9466587). Nevertheless, we modified the discussion to account for the possibility of defects in MT-dependent processes:

The nature of the statistical test is only described for Fig. 5C. The authors should clarify the specific statistical tests used for each subpanel where statistics were applied to ensure transparency and reproducibility.

All statistical tests unless otherwise stated were conducted using Student’s T-test.

Reviewer #2:

Enhancing the effectiveness of producing platelets from megakaryocytes (MKs) in vitro presents a major challenge and is of great importance. In this manuscript, Nakamura et al. performed high-throughput screening and identified microtubule (MT) destabilizing agents, including the vinca alkaloid vinblastine and vincristine (VCR), as potential promoters of imMKCL maturation and proplatelet formation. The overall study is interesting and significant for improving platelet yield from MKs. However, additional evidence is necessary to substantiate the claims made by the authors.

Major concerns

1. The authors found that imMKCLs sampled from culture systems with turbulent flow (turbulence+) showed reduced microtubule staining in terminally maturing imMKCLs compared to those sampled from static culture conditions (turbulence-), especially after day 3 (Fig 3D), similar to the expected phenotype of administering VCR. And the platelet yield per MK from the turbulence+ group is much higher than turbulence- group. How does the platelet yield per MK compare between the VCR treatment and the turbulence+ group? Can chemical compounds effectively replace the complex bioreactor devices?

Response: Reviewer 1 raised a similar point. VCR still has a platelet-biogenesis boosting effect in the context of differentiation under turbulent flow conditions (Fig. 4A). However, the effect size (fold-increase in platelet yields) is lower when VCR is used in the context of turbulent flow compared to static conditions (Fig 2D). We have added a plot (Fig 3E) and expanded the Discussion section to to provide a more detailed explanation of these findings and clarify their interpretation:

2. The authors compared surface-expression of P-Selectin/CD62P and PAC1 between control platelets and platelets activated with the platelet agonists ADP and TRAP-6. Since platelet are easily activated during the culturing and experimental processing, please provide the data for P-Selectin/CD62P and PAC1 expression without stimulation by any agonist.

Response: The missing control data is now included in revised figures 4A,B and S Fig 4A,B.

3. After transfused into mice, the circulation of VCR-iPSC-PLTs from both VCR treatment groups was significantly decreased compared to control iPSC-PLTs at all time points up to 24 hours. What is the possible mechanism underlying this phenomenon?

Response: Unfortunately, we do not know the mechanisms responsible for decreased platelet counts. We have modified and expanded the discuss of this issue in the Discussion section and acknowledge the value of future studies to address this question:

4. Although exposing imMKCLs to VCR boosts the yield of iPSC-PLTs, but VCR-iPSC-PLTs show reduced persistence and perform not as well as contr

---

## [Decision Letter · Decision Letter 1]

Association of Microtubule Destabilization With Platelet Yields in Terminally Differentiating hiPSC-derived Megakaryocyte Lines

PONE-D-24-54484R1

Dear Dr. Schlaeger,

We’re pleased to inform you that your manuscript has been judged scientifically suitable for publication and will be formally accepted for publication once it meets all outstanding technical requirements.

Kind regards,

Francesco Bertolini, MD, PhD

Academic Editor

PLOS ONE

Additional Editor Comments (optional):

Reviewers' comments:

Reviewer's Responses to Questions

**Comments to the Author**

Reviewer #1: All comments have been addressed

Reviewer #3: All comments have been addressed

2. Is the manuscript technically sound, and do the data support the conclusions?

Reviewer #1: Yes

Reviewer #3: Yes

3. Has the statistical analysis been performed appropriately and rigorously?

Reviewer #1: Yes

Reviewer #3: Yes

4. Have the authors made all data underlying the findings in their manuscript fully available?

Reviewer #1: Yes

Reviewer #3: Yes

5. Is the manuscript presented in an intelligible fashion and written in standard English?

Reviewer #1: Yes

Reviewer #3: Yes

Reviewer #1: The authors have done a thorough and commendable job in addressing the concerns raised during the first round of review. All comments have been adequately addressed, and the revised manuscript has been significantly improved. This reviewer has no further concerns.

Reviewer #3: The authors have adequately addressed the reviewers’ concerns. The paper is now suitable for publication.

**Do you want your identity to be public for this peer review?** For information about this choice, including consent withdrawal, please see our Privacy Policy

Reviewer #1: No

Reviewer #3: No

---

## [Editor Report · Acceptance letter]

PONE-D-24-54484R1

PLOS ONE

Dear Dr. Schlaeger,

I'm pleased to inform you that your manuscript has been deemed suitable for publication in PLOS ONE. Congratulations! Your manuscript is now being handed over to our production team.

Kind regards,

on behalf of

Dr. Francesco Bertolini

Academic Editor

PLOS ONE